# TextMixer: Mixing Multiple Inputs for Privacy-Preserving Inference

**Xin Zhou**[1*], **Yi Lu**[1*], **Ruotian Ma**[1], **Tao Gui**[2†], **Qi Zhang**[1†], **Xuanjing Huang**[1,3]

[1]School of Computer Science, Fudan University, Shanghai, China
[2] Institute of Modern Languages and Linguistics, Fudan University, Shanghai, China
[3] International Human Phenome Institutes, Shanghai, China
{xzhou20, tgui, qz}@fudan.edu.cn, yilu23@m.fudan.edu.cn

## Abstract

Pre-trained language models (PLMs) are often deployed as cloud services, enabling users to upload textual data and perform inference remotely. However, users' personal text often contains sensitive information, and sharing such data directly with the service providers can lead to serious privacy leakage. To address this problem, we introduce a novel privacy-preserving inference framework called ***TextMixer***, which prevents plaintext leakage during the inference phase. Inspired by $k$-anonymity, TextMixer aims to obfuscate a user's private input by mixing it with multiple other inputs, thereby confounding potential privacy attackers. To achieve this, our approach involves: (1) proposing a novel encryption module, Privacy Mixer, which encrypts input from three distinct dimensions: mixing, representation, and position. (2) adopting a pre-trained Multi-input Multi-output network to handle mixed representations and obtain multiple predictions. (3) employing a Privacy Demixer to ensure only the user can decrypt the real output among the multiple predictions. Furthermore, we explore different ways to automatically generate synthetic inputs required for mixing. Experimental results on token and sentence classification tasks demonstrate that TextMixer greatly surpasses existing privacy-preserving methods in both performance and privacy.

## 1 Introduction

The emergence of pre-trained language models (PLMs) (Devlin et al., 2018; Liu et al., 2019; Brown et al., 2020) has significantly increased the demand for cloud inference services. By uploading their textual inputs to the cloud, users can benefit from PLMs and achieve superior performance on various

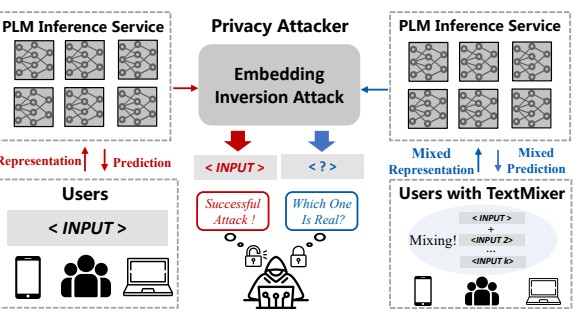

Figure 1: Comparison of the baseline (left) with our method (right). Without privacy protection, word representations can be easily restored to plain text. Our method utilizes mixing to conceal the real input among multiple inputs, thereby confusing privacy attackers and preserving the plain text privacy during inference phase.

NLP tasks, without requiring high-performance hardware or expertise. However, the widespread adoption of PLM services also introduces a risk of privacy leakage (Jegorova et al., 2022). Users' textual inputs may contain sensitive information, including names, schedules, and even business secrets. Leaking such private information to service providers is unacceptable to most users.

To address users' privacy concerns and comply with relevant laws[1], service providers can provide privacy-preserving inference services. Many sought to make users **query PLM services with word representations rather than plaintext**. But digital word representations still need additional privacy protection because recent studies (Pan et al., 2020; Song and Raghunathan, 2020) have revealed that standard word representation can be easily restored to its original word under embedding inversion attacks (Höhmann et al., 2021).

One solution is to apply cryptographic techniques to the PLM (Hao et al.; Li et al., 2022; Zheng et al., 2023; Chen et al., 2022), but these methods often come with significant communi-

---

*Equal contribution.
†Corresponding authors.

[1]https://www.consilium.europa.eu/en/policies/data-protection/data-protection-regulation/

cation costs and computational time, making it difficult to apply them in real-world scenarios. Another potential solution is to remove the private information in word representations (Pan et al., 2020) through adversarial training (Li et al., 2018; Coavoux et al., 2018; Plant et al., 2021) and differential privacy (Lyu et al., 2020a; Hoory et al., 2021; Yue et al., 2021). However, the private information in our scenario pertains to each word in the user's plaintext. **Removing the word information from word representations is highly unnatural and often leads to crucial performance degradation** (Zhou et al., 2022).

In this paper, we propose a novel privacy-preserving inference paradigm named TextMixer. Our method is inspired by $k$-anonymity (Sweeney, 2002), a classic privacy-preserving technique that ensures each individual's information cannot be distinguished from at least $k$-1 other individuals. For that purpose, **TextMixer aims to make each real word representation indistinguishable by mixing it with multiple synthetic representations during the inference phase**, rather than removing word information, as shown in Figure 1.

However, directly mixing the representations of $k$ inputs leads to information interference, and normal PLMs cannot handle the mixed inputs to acquire the desired output. To address the above problems, we resort to Multi-input Multi-output (MIMO) network (Ramé et al., 2021; Murahari et al., 2022, 2023), allows us to process mixed representations with minimal information interference and obtain multiple predictions simultaneously. We further design a privacy mixer to enhance TextMixer's privacy, which encrypts the real inputs from three dimensions. For each forward pass, the privacy mixer applies **anonymous mixing, representation perturbation, and position shuffling** to encrypt the real inputs. Only the user holds the correct keys to decrypt the real prediction from mixed representation. In this way, TextMixer preserves the privacy of both the input and prediction. We also conduct an investigation into different methods for generating synthetic inputs. Our contribution can be summarized as follows[2]:

- We propose TextMixer, a privacy-preserving inference method that obfuscates privacy attacks by mixing private input with synthetic inputs.

- We propose a mixing-centric encryption method and explore several approaches for synthetic input generation.

- Experimental results on various token and sentence classification datasets demonstrate TextMixer's effectiveness in privacy and performance.

## 2 Methodology

In this section, we present a novel privacy-preserving inference framework TextMixer, which achieves $k$-anonymity (Sweeney, 2002) by mixing multiple inputs during the inference phase.

### 2.1 Overview

We take an example to show how $k$-anonymity contributes to privacy-preserving inference. To hide the private text "John", TextMixer mixes the word representation of "John" with $k$-1 synthetic ones (such as "Mike", "Smith", "Johnson", etc.). Only the mixed representation is shared with privacy attackers, making it difficult for privacy attackers to identify which word is real. As shown in Figure 2, our method consists of three steps: (1) ***User*** first generates $k$-1 synthetic representations (§2.5) and inputs $k$ representations into the Privacy Mixer (§2.3), which uses various encryption methods with mixing as the core to combine the real and synthetic inputs, creating mixed and encrypted representations; (2) ***Cloud server*** receives the mixed representations, and inputs them into a pre-trained MIMO network (§2.2), which can handle mixed representations formed by multiple inputs and output their outputs; (3) ***User*** uses a Privacy DeMixer (§2.4) to decrypt the desired prediction from the multiple outputs locally. In the following subsection, we provide a detailed introduction to each part of TextMixer. **We also conduct a theoretical privacy analysis of TextMixer in Appendix B.**

### 2.2 MIMO Network

We first adopt the mixing process of MIMO Network (Murahari et al., 2023) (called **vanilla mixing** here) to our scenarios. Vanilla mixing transforms different inputs into separate spaces before mixing, thus reducing representation interference between multiple inputs.

Given a user's real input sequence $\mathbf{X}^r = [\mathbf{x}_1^r, ..., \mathbf{x}_n^r]$ with $n$ word representations, and $k-1$ synthetic inputs $\{\mathbf{X}^{s_i}\}_{i=1}^{k-1}$ where $\mathbf{X}^{s_i} =$

---

[2]We release our code at https://github.com/LuLuLuyi/TextMixer

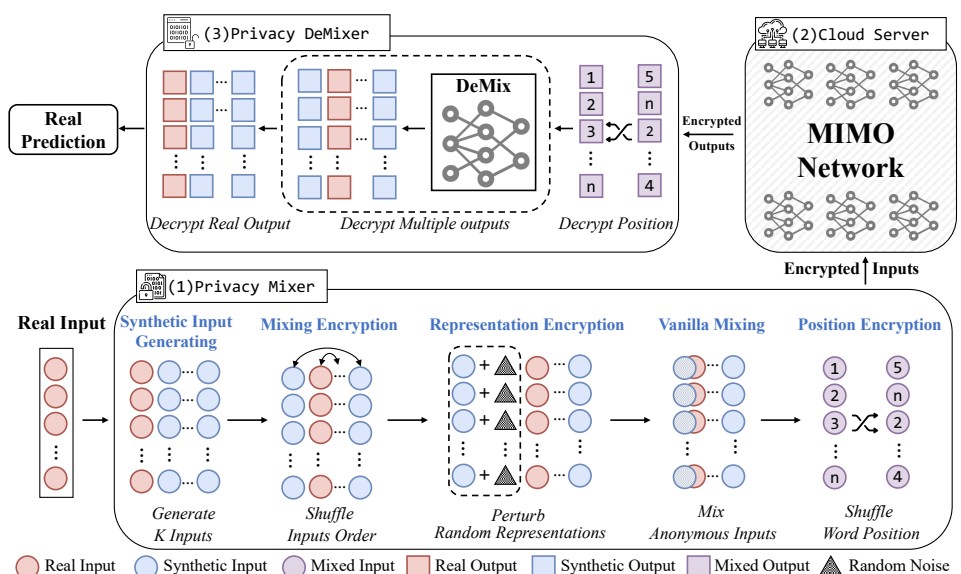

Figure 2: An overview of TextMixer's inference. Privacy Mixer first encrypts the real input locally on the user's device, obtaining the encrypted input through three aspects: anonymous mixing, perturb representation and random position. The MIMO network deployed in the cloud only received encrypted inputs for subsequent inference and returns the encrypted output back to the user. With the help of the user's private keys, Privacy DeMixer can decrypt the real prediction from the encrypted output.

$[\mathbf{x}_1^{s_i}, ..., \mathbf{x}_n^{s_i}]$, we aims to mix these representation to generate $\mathbf{X}^{mix} = [\mathbf{x}_1^{mix}, ..., \mathbf{x}_n^{mix}]$. For simplicity, we denote $k$ words to be mixed at the $i$-th position as $\mathbf{x}_i^{1:k} = [\mathbf{x}_i^r, \mathbf{x}_i^{s_1}, ..., \mathbf{x}_i^{s_{k-1}}]$. To avoid misunderstanding, **we use "position" to represent the word position in an input sequence, and "order" to represent the order of $k$ inputs**. For example, $\mathbf{x}_i^j$ means the word representation at $i$-th position from $j$-th input. To ensure that the mixed representation still retains the individual semantics of each input, we need to perform transformation before mixing. Transformation maps word representations from different inputs to separate spaces and consequently reduces interference between their representations. The process of transformation and mixing can be defined as:

$$\mathbf{x}_i^{mix} = \frac{1}{k} \sum_{j=1}^{k} \mathbf{x}_i^j \odot \mathbf{e}^j, \qquad (1)$$

where $\mathbf{x}_i^j$ is $j$-th representation in $\mathbf{x}_i^{1:k}$ and $\mathbf{e}^j \in \mathbb{R}^d \sim \mathcal{N}(\mathbf{0}, \mathbf{I})$ is transformation vector sampled from a standard multivariate Gaussian distribution.

Next, we input mixed representations into a MIMO network to obtain multiple predictions. Then we demix the real prediction from the multiple predictions, as shown in §2.4. The structure of MIMO network is the same as BERT, its ability to handle mixed representation comes

from the over-parameterized network and MIMO pre-training (Murahari et al., 2023).

## 2.3 Privacy Mixer

**However, vanilla mixing process is not a good privacy protector due to the lack of randomness.** For example, equation 1 shows that different inputs will be mapped to different spaces. If the real input is always mapped to a fixed space, privacy attackers can easily identify this fixed pattern, thereby restoring the real input from the mixed representation (shown in Table 2). Thus, we design privacy mixer, which extends the vanilla mixing into a real encryption strategy from three dimensions: **anonymous mixing**, **perturbed representation**, and **random position**.

**Mixing Encryption.** Inspired by $k$-anonymity, this encryption aims to make each real representation $\mathbf{x}_i^r$ indistinguishable from synthetic ones $\{\mathbf{x}_i^{s_j}\}_{j=1}^{k-1}$. As mentioned before, the real input cannot be mapped to a fixed space, **thus we shuffle inputs to get a random mixing order**, allowing the real input to be anonymous:

$$\mathbf{R}_{ME} \cdot \mathbf{x}_i^{1:k} = [\mathbf{x}_i^{s_{k-1}}, ..., \mathbf{x}_i^r, \mathbf{x}_i^{s_1}], \qquad (2)$$

where $\mathbf{x}_i^{1:k} \in \mathbb{R}^{k \times d}$ and $\mathbf{R}_{ME} \in \{0, 1\}^{k \times k}$ is a random permutation matrix. Equation 2 represents only one possible shuffling result. $\mathbf{R}_{ME}$

**is generated by the user's side and not shared with the cloud.**

**Representation Encryption.** To obscure the representation itself, we add random perturbations to each representation of a random input, as shown in Figure 2. Following Plant et al. (2021), we adopt Laplace Noise as the random perturbation. Formally, the process of mixing encryption and representation encryption for $\mathbf{x}_i^{1:k}$ is:

$$\mathbf{x}_i^{Enc(1:k)} = \mathbf{R}_{ME} \cdot [\mathbf{x}_i^r, \mathbf{x}_i^{s_1}, ..., \mathbf{x}_i^{s_\alpha} + Lap(\epsilon), ..., \mathbf{x}_i^{s_{k-1}}], \quad (3)$$

where $\epsilon$ is a hyperparameter controlling the scale of noise, smaller values of correspond to a larger scale of noise. $Lap(\epsilon)$ can be added to any representation, including real input $\mathbf{x}_i^r$.

Then we mix $k$ representation $\mathbf{x}_i^{Enc(1:k)}$ at each position $i$, according to equation 1. Finally, we obtain the mixed representations $\mathbf{X}^{mix} \in \mathbb{R}^{n \times d}$.

**Position Encryption.** To further enhance the privacy of $\mathbf{X}^{mix}$, position encryption aims to reduce its human readability. We utilize a random permutation matrix, denoted as $\mathbf{R}_{PE} \in \{0, 1\}^{n \times n}$, to shuffle word positions in $\mathbf{X}^{mix}$:

$$\mathbf{X}_{PE}^{mix} = \mathbf{R}_{PE} \cdot \mathbf{X}^{mix} = [\mathbf{x}_n^{mix}, \mathbf{x}_1^{mix}..., \mathbf{x}_{n-1}^{mix}]. \quad (4)$$

In this way, even if privacy attackers can restore some text from mixed representation, they cannot obtain a readable sequence. We show that $\mathbf{X}_{PE}^{mix}$ and $\mathbf{X}^{mix}$ are equivalent for PLMs in Appendix D. **Similar to Mixing Encryption, only the user holds the private key $\mathbf{R}_{PE}$.**

Finally, users upload encrypted representations, $\mathbf{X}_{PE}^{mix}$, to the cloud for further computation. A pre-trained MIMO network processes the $\mathbf{X}_{PE}^{mix}$ and sends back the final representation $\mathbf{H}_{PE}^{mix} \in \mathbb{R}^{n \times d}$ to the user's side for de-mixing the real prediction.

### 2.4 Privacy DeMixer

The goal of Privacy DeMixer is to decrypt the encrypted final representations $\mathbf{H}_{PE}^{mix}$ and obtain the desired prediction, with the help of **user's personal keys $\mathbf{R}_{PE}$ and $\mathbf{R}_{ME}$.** First, we use $\mathbf{R}_{PE}$ to restore original word position:

$$\mathbf{H}^{mix} = \mathbf{R}_{PE}^{-1} \cdot \mathbf{H}_{PE}^{mix}, \quad (5)$$

where $\mathbf{H}^{mix} = [\mathbf{h}_1^{mix}, ..., \mathbf{h}_n^{mix}]$. Next, we de-mix these mixed final representations $\mathbf{H}^{mix}$ to get final representations corresponding to different inputs.

For simplicity, we only consider the representation in position $i$. De-mixing means extracting $k$ final representations $\mathbf{h}_i^{1:k} \in \mathbb{R}^{k \times d}$ from $\mathbf{h}_i^{mix} \in \mathbb{R}^d$, where $\mathbf{h}_i^{1:k}$ are the final representations of different inputs $\mathbf{x}_i^{1:k}$. The final representation of $j$-th input in $\mathbf{x}_i^{1:k}$ can be obtained by:

$$\mathbf{h}_i^j = \text{DeMix}\left([\mathbf{h}_i^{mix}; \mathbf{d}^j]\right), \quad (6)$$

where DeMix is a two layer MLP and $\mathbf{d}^j \in \mathbb{R}^d$ is private vectors to de-mix the $j$-th representation from mixed representation and $[*; *]$ means concatenation. The de-mixing ability of DeMix and $\mathbf{d}^j$ comes from pre-training (Murahari et al., 2023). We show that DeMix and $\mathbf{d}^j$ are not helpful in restoring the real input in §4.3.

After de-mixing, we restore an ordered set $\mathbf{h}_i^{1:k} = [\mathbf{h}_i^1, ..., \mathbf{h}_i^k]$, then we can use $\mathbf{R}_{ME}$ to decrypt the original order by $\mathbf{R}_{ME}^{-1} \cdot \mathbf{h}_i^{1:k}$, obtaining the real input's final representation $\mathbf{H}^r \in \mathbb{R}^{n \times d}$. Subsequently, $\mathbf{H}^r$ is fed into the classification layer to obtain the prediction desired by the user.

### 2.5 Synthetic Input Generating

As mentioned before, our method requires $k$-1 synthetic inputs for mixing. We now explore principled ways of generating synthetic input.

**Generating with Real Data.** The most intuitive method is to select the real training data as the synthetic data. Given a real input sentence, we randomly sample $k - 1$ sentences from the training set and encrypt these sentences.

**Generating with Similarity.** In this approach, we leverage the similarity for synthetic input generating. Given a real word $\mathbf{x}_r$, we select the closest word $w_s$ in the embedding space as its synthetic word:

$$w_s = \arg\max_{k-1} ||\mathbf{w} - \mathbf{x}_r||, \quad (7)$$

where $\mathbf{w}$ is the embedding matrix of PLM.

**Generating with Input Itself.** This method is inspired by Zhou et al. (2022) that fuses representation within the same sentence. Given a real input $\mathbf{X}^r = [\mathbf{x}_1^r, ..., \mathbf{x}_n^r]$, for each $\mathbf{x}_i^r$, we random sample $k - 1$ representations from $\mathbf{X}^r$ (excluding $\mathbf{x}_i^r$ itself) as the synthetic inputs.

### 2.6 Training Details

TextMixer supports both token-level and sentence-level classification tasks. During the training

process, we randomly sample k inputs from the training set and only enable representation encryption. We add random noise to a random input sequence and then mix all the inputs together. The mixed input is sent to a pre-trained MIMO network to get the final representation. Without any decryption process, we directly use DeMix to obtain the final representation of each input, which is then fed into the output layer to predict the task label. In addition to the task loss, a retrieval loss is added to promote the network's ability to handle multiple inputs (Murahari et al., 2023). The final loss can be represented as follows:

$$\mathcal{L}_{retrieval}(x^{1:k}) = \sum_i -logP(w_i^j|\mathbf{h}_i^j), \quad (8)$$

$$\mathcal{L} = \mathcal{L}_{task} + \alpha\mathcal{L}_{retrieval}, \quad (9)$$

where $\alpha$ is a hyperparameter, $\mathbf{h}_i^j$ is final representation of the $j$-th sentence at position $i$, and $w_i^j$ is the word corresponding to $\mathbf{h}_i^j$. During the inference phase, we can directly enable all encryption strategies and switch the synthetic input generation methods without retraining.

## 3 Experimental Setup

### 3.1 Datasets

We conduct experiments on four representative datasets. **Sentence classification**: we select IMDB (Maas et al., 2011) for sentiment analysis and AGNews (Zhang et al., 2015) for topic classification. **Token classification**: we select NER (Named Entity Recognition) as the main task, including CoNLL2003 (Tjong Kim Sang and De Meulder, 2003) and Ontonotes (Weischedel et al., 2013). These tasks are highly relevant to real-world scenarios and datasets often involve sensitive information such as names and locations. Thus they serve as suitable benchmarks for evaluating both privacy and performance. The details of these datasets are in Appendix A.

### 3.2 Baselines

For a thorough comparison, we select the widely used privacy attack and defense baselines.

**Attack Baselines.** Following Zhou et al. (2022), we select three privacy attack baselines. *KNN* (Qu et al., 2021) selects the closest word in the embedding space as the real word. *InvBert* (Höhmann et al., 2021) trains an embedding inversion model that takes word representations

as input and outputs each representation's original word one-to-one. *MLC* (Song and Raghunathan, 2020) is similar to InvBert, the difference is that MLC performs multi-label classification tasks, predicting all the words that have appeared without emphasizing the order.

**Defense Baselines.** *DPNR* (Lyu et al., 2020b) uses differential privacy to obfuscate representations. *CAPE* (Plant et al., 2021) further eliminates privacy information by incorporating adversarial training and differential privacy. *SanText+* (Yue et al., 2021) uses differential privacy and word frequency to replace sensitive words. *TextFusion* (Zhou et al., 2022) employs word fusion and misleading training to hinder privacy attackers from training targeted embedding inversion models. We attack it with a special InvBert due to different settings, as shown in Appendix C. *DataMix* (Liu et al., 2020) proposes to mix and encrypt the image with random coefficients.

### 3.3 Evaluation Metrics

We consider privacy metrics at both the word level and the sentence level. At the word level, we use the **Top-k**, which refers to the proportion of real words among the top $k$ predictions generated by the attack methods for a given representation. At the sentence level, we use the **RougeL** (Lin, 2004), a text generation metric, to measure the similarity between the restored sentence and the original sentence. For MLC Attack, we use the **Token-Hit** (Zhou et al., 2022), which treats the words in real input as a set, calculating the percentage of predicted words in the raw words.

### 3.4 Implementation Details

We show three important implementation details below, and present the remaining details in Appendix G. (1) **All attack and defense methods are performed at the embedding layer, before any transformer modules**. Users only need to convert words into embeddings and send them to the cloud. (2) The pre-trained MIMO network is based on BERT-base, thus we also chose BERT-base as the backbone to train task models for all baselines, as well as the inversion models for InvBert and MLC. (3) Inversion models for each dataset are trained in its training set.

| Dataset | Method | Task ↑ | KNN ↓ | | | InvBert ↓ | | | MLC ↓ |
|---|---|---|---|---|---|---|---|---|---|
| | | | Top-1 | Top-5 | Rouge | Top-1 | Top-5 | Rouge | Token-Hit |
| IMDB | Fine-tuning (Devlin et al., 2018) | 94.10 | 100 | 100 | 91.70 | 100 | 100 | 99.28 | 51.13 |
| | DPNR (Lyu et al., 2020b) | 85.00 | 0.11 | 58.42 | 0.08 | 99.82 | 99.99 | 99.17 | 43.01 |
| | CAPE (Plant et al., 2021) | 85.88 | 5.70 | 71.78 | 4.12 | 100 | 100 | 99.28 | 48.05 |
| | SanText+ (Yue et al., 2021) | 50.16 | 69.21 | 69.27 | 59.19 | 73.94 | 81.24 | 73.47 | 42.48 |
| | DataMix (Liu et al., 2020) | 50.36 | 29.51 | 76.08 | 26.57 | 66.48 | 98.20 | 60.55 | 28.13 |
| | TextFusion (Zhou et al., 2022) | 90.60 | 39.46 | 85.30 | 46.17 | 48.93 | 91.94 | 61.38 | 54.43 |
| | **TextMixer (Ours)** | **90.93** | **0.00** | **0.00** | **0.00** | **12.17** | **36.07** | **0.35** | **21.11** |
| AGNews | Fine-tuning (Devlin et al., 2018) | 94.92 | 100 | 100 | 93.66 | 100 | 100 | 99.81 | 89.31 |
| | DPNR (Lyu et al., 2020b) | 90.01 | 4.11 | 24.35 | 3.64 | 28.50 | 45.05 | 28.94 | 27.22 |
| | CAPE (Plant et al., 2021) | 90.22 | 0.36 | 1.18 | 0.36 | 5.73 | 10.72 | 5.91 | 22.13 |
| | SanText+ (Yue et al., 2021) | 90.80 | 65.24 | 65.28 | 52.90 | 70.11 | 76.94 | 70.28 | 53.91 |
| | DataMix (Liu et al., 2020) | 91.62 | 59.19 | 98.41 | 55.68 | 95.26 | 99.85 | 89.14 | 34.94 |
| | TextFusion (Zhou et al., 2022) | 91.89 | 7.21 | 26.80 | 4.22 | 49.03 | 87.69 | 59.61 | 71.01 |
| | **TextMixer (Ours)** | **92.23** | **0.00** | **0.00** | **0.00** | **2.22** | **5.62** | **0.10** | **16.09** |
| CoNLL2003 | Fine-tuning (Devlin et al., 2018) | 90.47 | 100 | 100 | 95.01 | 100 | 100 | 99.83 | 61.47 |
| | DPNR (Lyu et al., 2020b) | 39.09 | 13.86 | 30.81 | 9.96 | 11.27 | 23.55 | 11.37 | 25.46 |
| | CAPE (Plant et al., 2021) | 70.53 | 0.11 | 5.90 | 0.06 | 37.44 | 45.28 | 37.91 | 27.56 |
| | SanText+ (Yue et al., 2021) | 72.19 | 76.56 | 76.85 | 55.24 | 75.62 | 82.02 | 75.74 | 40.93 |
| | DataMix (Liu et al., 2020) | 2.38 | 63.69 | 97.82 | 60.96 | 78.98 | 97.24 | 74.80 | 39.32 |
| | TextFusion (Zhou et al., 2022) | 77.53 | 4.91 | 21.40 | 2.64 | 22.27 | 27.02 | 49.24 | 55.09 |
| | **TextMixer (Ours)** | **86.24** | **0.00** | **0.00** | **0.00** | **4.14** | **11.26** | **0.93** | **18.45** |
| Ontonotes 5.0 | Fine-tuning (Devlin et al., 2018) | 87.07 | 100 | 100 | 95.01 | 100 | 100 | 99.82 | 81.11 |
| | DPNR (Lyu et al., 2020b) | 42.10 | 10.64 | 41.24 | 9.26 | 24.84 | 42.85 | 25.22 | 31.17 |
| | CAPE (Plant et al., 2021) | 71.23 | 1.97 | 11.42 | 1.51 | 22.66 | 29.10 | 22.77 | 30.68 |
| | SanText+ (Yue et al., 2021) | 65.32 | 73.66 | 73.82 | 55.89 | 73.22 | 80.69 | 73.36 | 48.61 |
| | DataMix (Liu et al., 2020) | 0.21 | 65.76 | 97.75 | 62.52 | 90.39 | 99.11 | 85.67 | 55.03 |
| | TextFusion (Zhou et al., 2022) | 79.30 | 3.86 | 26.13 | 0.65 | 21.31 | **25.35** | 39.81 | 73.89 |
| | **TextMixer (Ours)** | **82.38** | **0.00** | **0.00** | **0.00** | **11.45** | 31.12 | **0.61** | **20.95** |

Table 1: Main results on four classification datasets. All attack and defense methods are carried out at the **embedding layer**. **Task** denotes the metric for the task performance, we report accuracy for IMDB and AGNEWS, F1 score for CoNLL2003 and Ontonotes5.0. Top-1, Top-5, Rouge, and Token-Hit are the privacy leakage metric. For task performance, higher is better; for privacy leakage metric, lower is better. The best performance and privacy in defense methods are highlighted in bold.

# 4 Experimental Results

In this section, we show the results of privacy and task performance in §4.1 and the ablation study in §4.2. We also design three specific privacy attacks for TextMixer and show the attack results in §4.3. The inference cost is shown in Appendix F.

## 4.1 Main Results

Main results on baselines and TextMixer are listed in Table 1. From the result, we can see that (1) Normal word representations without any defense method suffer from privacy leakage risk. Almost all privacy attacks achieve a 100% success rate in fine-tuning, which means that privacy is completely leaked to attackers. **(2) Preserving privacy in the embedding layer is an extremely challenging task.** Despite compromising significant performance, most baselines are unable to provide satisfactory privacy protection. This is due to the inherent contradiction of removing private word information from word representations, as discussed in the introduction. (3) Our proposed TextMixer achieves better task performance and

lower privacy leakage than most baselines across all datasets. With the help of our proposed encryption strategy and MIMO network, TextMixer avoids contradictory behavior, thus achieving good performance while preserving privacy.

Previous works usually require deploying multiple transformer layers in users' devices to achieve satisfying performance. But when applied to the embedding layer, we observe that most baselines are difficult to provide privacy protection. CAPE and DPNR usually need to trade a lot of task performance for the ability to protect privacy, which can be attributed to the inherent contradiction that we discussed above. SanText+ relies on word replacement to protect privacy, but word replacement cannot handle token-level tasks, which require much token information, and tasks with long sequences, which require lots of words to be replaced, such as IMDB. TextFusion fuses most words to reduce Top-1, resulting in serious performance degradation. But the performance of Top-5 and MLC of TextFusion remains high, indicating privacy risk is still high under our strict

privacy setting. DataMix fails to perform well on most datasets, and we believe there are two reasons. First, its encryption mechanism is friendly to convolutional networks but not suitable for PLMs that involve more non-linear operations. Second, DataMix is designed for images, whereas we require encrypting a representation sequence, which is a harder task. These two difficulties cause it to fail on PLMs and textual data.

## 4.2 Ablation Study

In this subsection, we explore the effect of encryption strategy and synthetic input generation. We also show the effect of mixed inputs number and noise scale in Appendix E.

**Effect of encryption strategy.** From Table 2, we can see that **mixing Encryption** plays an important role in privacy. Without mixing encryption, privacy attackers can easily identify the space of real input and accurately restore the original words, resulting in high privacy leakage. **Representation Encryption** prevents privacy attackers by adding noise to representation. Similar to the previous work, adding noise hurts task performance, but an appropriate noise would not result in much performance loss. This is acceptable as it contributes largely to privacy. **Position Encryption** is not designed to prevent attackers from restoring words, but it can significantly reduce readability, resulting in a substantial decrease in Rouge scores. From the above results, it can be observed that our proposed encryption techniques reduce privacy leakage risks from different perspectives while having minimal impact on performance.

| Dataset | Method | Task | KNN | | InvBert | |
|---|---|---|---|---|---|---|
| | | | Top-1 | Rouge | Top-1 | Rouge |
| AGNews | TextMixer | 92.32 | **0.00** | **0.00** | **2.22** | **0.10** |
| | - Mix. | 91.73 | 0.00 | 0.00 | 89.22 | 2.07 |
| | - Repr. | **92.63** | 0.00 | 0.00 | 6.57 | 0.13 |
| | - Pos. | 92.32 | 0.00 | 0.00 | 2.22 | 2.19 |
| CoNLL03 | TextMixer | 86.24 | **0.00** | **0.00** | **4.14** | **0.93** |
| | - Mix. | 86.64 | 0.00 | 0.00 | 88.64 | 5.89 |
| | - Repr. | **86.69** | 0.00 | 0.00 | 15.53 | 1.34 |
| | - Pos. | 86.24 | 0.00 | 0.00 | 4.14 | 4.20 |

Table 2: Effect of different encryption strategies. Mix. means Mixing Encryption, Repr. means Representation Encryption and Pos. means Position Encryption.

**Effect of Synthetic Inputs.** From Figure 3, we surprisingly find that the similarity-based methods do not confuse the inversion model but instead lead to more severe privacy leakage. We speculate

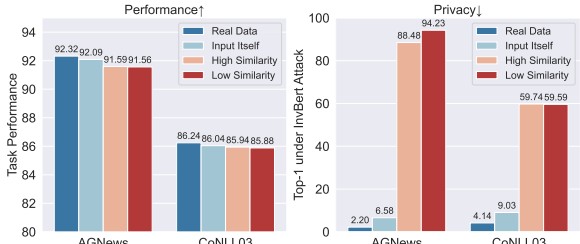

Figure 3: Effect of different synthetic inputs. Privacy metric is Top-1 under InvBert attack. Performance metric is accuracy for AGNews and F1 for CoNLL03.

that these synthetic inputs differ significantly from the words that would appear in the real world. During the training process, the powerful inversion model gradually eliminates this irrelevant information and identifies the feature of the real input, resulting in a successful attack. Real Data and Input Itself achieve first and second rankings in preserving privacy. Both of these synthetic inputs are generated based on real data, which enlightens us that the closer synthetic inputs approximate real-world scenarios, the greater their potential to mislead privacy attackers. The substitution of synthetic inputs has only a marginal impact on performance, both in token-level tasks and sentence-level tasks, indicating users can replace the synthetic inputs generation method at any time without the need for retraining.

## 4.3 Stronger Privacy Attacks

Although TextMixer can defend the powerful embedding inversion attack, there are still concerns about potential privacy leakage under stronger privacy attacks. Therefore, we design some attacks for our encryption method.

| Dataset | PIA | DMA | | |
|---|---|---|---|---|
| | Acc. | Top-1 | Top-5 | Rouge |
| AGNews | 53.64 | 0.00 | 0.00 | 0.00 |
| CoNLL03 | 48.09 | 0.00 | 0.43 | 0.00 |

Table 3: Results of DeMixing Attack (DMA) and Position Inversion Attack (PIA).

**DeMixing Attack** is designed for Mixing Encryption. We verify whether DeMix and $\mathbf{d}^j$, which are used to demix the outputs, are helpful to demix the mixed input. We assume the privacy attacker knows the exact order of the real input. For instance, they know the $j$-th input is the real one, thus they can use DeMix and $\mathbf{d}^j$ to demix the mixed input and obtain the word representations of $j$-th input. Then

| Train\Test | Real Data | Input Itself | High Similarity | Low Similarity |
|---|---|---|---|---|
| Real Data | 4.14 | 4.21 | 21.74 | 0.19 |
| Input Itself | 3.48 | 9.03 | 23.09 | 0.02 |
| High Similarity | 4.13 | 5.18 | 59.74 | 0.00 |
| Low Similarity | 0.60 | 0.17 | 0.86 | 59.50 |

Table 4: Results of Cross Inversion Attack on CoNLL03. Columns represent the training data for InvBert, while the rows represent the testing data.

they use an InvBert to predict real words and train DeMix, $\mathbf{d}^j$ and InvBert jointly. However, attack results (DMA) shown in Table 3 show that **DeMix is not helpful for restoring real input**, this attack performs even worse than using InvBert directly. We speculate that this is because DeMix is trained to demix the output instead of input, it strongly interferes with the inversion model, making it impossible to attack successfully at all.

**Position Inversion Attack** is designed for Position Encryption. It aims to restore the original word position from the encrypted input. Similar to InvBert, we train a model to predict the original position of each encrypted representation and show attack results (PIA) in Table 3. The accuracy of restoring the original position is only 50%, and the readability of such a sentence remains low. Within TextMixer, it is not a threat since embedding inversion attacks can only restore a few words from encrypted representations.

**Cross Inversion Attack** is designed for synthetic input generation, which verifies whether the InvBert trained on one synthetic input generation method works better on another one. From Table 4, we can see that all transferred InvBerts perform worse than InvBert trained and tested in the same synthetic input indicating its poor generalization ability. Privacy attacks' performance primarily depends on the synthetic input used during testing.

## 5 Related Work

### 5.1 Plaintext Privacy during Inference

**Privacy Threats.** Most PLM services require users to upload their plaintext data to the cloud to perform PLM inference, which leads to a serious risk of privacy leakage (Jegorova et al., 2022). Privacy attackers could be the service provider themselves, they aim to collect users' textual input for their own purposes. For example, they can use these data to train a better model or extract users' private information, such as personal details and confidential business information, even if it is prohibited by law. Recent literature (Song

and Shmatikov, 2019; Pan et al., 2020) shows that even uploading word representations can still leak privacy, as the embedding inversion attack (Höhmann et al., 2021) can restore word representations to their original words.

**Privacy-preserving Methods.** To avoid users' private plaintext from leaking to privacy attackers, many existing works adopt cryptographic techniques such as Homomorphic Encryption (Gentry, 2009; Chen et al., 2022) and Secure Multiparty Computation (Evans et al., 2018; Li et al., 2022) into PLM (Hao et al.; Zheng et al., 2023). Although theoretically guaranteed, these methods typically require much higher time and communication costs than standard PLM. For example, Hao et al. shows that an encrypted BERT-tiny model requires about 50 seconds and 2GB of communication cost to process a 30 words sentence. An alternative that does not require additional inference costs is to employ adversarial training (Li et al., 2018; Coavoux et al., 2018; Plant et al., 2021) and differential privacy (Lyu et al., 2020a; Hoory et al., 2021; Yue et al., 2021) to remove private information in representations. However, the perturbed word representations, which lack sufficient information for privacy attackers to identify their original words, are also insufficient for the model to achieve high performance on downstream tasks (Zhou et al., 2022). Therefore, in this work, we propose mixing inputs rather than removing word information to preserve plaintext privacy during the inference phase.

**Comparison with Similar Works.** Huang et al. (2020) propose TextHide, which mixes up inputs during the training phase to protect the privacy of training data. Our work focuses on users' plaintext privacy and mixes inputs during the inference phase, which brings new challenges. Liu et al. (2020) propose Datamix to mix multiple inputs during inference and encrypt them with random coefficients. Our encryption method is different from Datamix, and Datamix is designed for image data and convolutional networks, which not performs well in text data and PLM. Recently, Zhou et al. (2022) propose TextFusion, which fuses parts of word representations within the same sentence to prevent privacy attackers from training a targeted inversion model. Differently, our method mixes representations from different sentences and our privacy setting is more challenging. We discuss

it in Appendix C.

## 5.2 Multi-input Multi-output Network

Multi-input Multi-output network aims to use one neural network to process multiple inputs and obtain their predictions in one forward pass. Havasi et al. (2020) and Ramé et al. (2021) try to improve robustness by ensembling multiple predictions in a MIMO convolutional network. Murahari et al. (2022) propose a transformer-based MIMO model for efficient inference and further improve its performance through pre-training (Murahari et al., 2023). Our work does not focus on robustness or efficiency but rather utilizes the MIMO network as a tool to construct partial encryption strategies, thereby achieving privacy-preserving inference.

## 6 Conclusion

In this paper, we propose a novel privacy-preserving inference framework TextMixer. Our framework is inspired by $k$-anonymity and makes real input indistinguishable by mixing multiple inputs during the inference phase. Specifically, TextMixer adopts a pre-trained MIMO network, which can process representation composed of multiple inputs and obtain multiple outputs simultaneously, to handle mixed representations. We further propose the privacy mixer to encrypt not only the mixing but also the two dimensions of representation and word position. We also explore various ways to automatically generate synthetic inputs. At the embedding layer, TextMixer outperforms the existing privacy-preserving methods by a large margin, regardless of token and sentence classification tasks. The detailed ablation and analysis experiments also show TextMixer is an effective privacy-preserving inference framework.

## Limitations

We have summarized three limitations of TextMixer. (1) TextMixer is designed to protect plaintext privacy, and we have only conducted experiments to evaluate its privacy under embedding inversion attacks. However, TextMixer provides the $k$-anonymity by mixing multiple inputs, it has the potential to address other privacy concerns, such as resisting Attribute Inference Attacks. (2) TextMixer's privacy guarantee is based on $k$-anonymity, which is not as rigorous as homomorphic encryption. Although TextMixer's

privacy has been empirically verified through the powerful and widely used embedding inversion attack, and we have provided privacy proof in Appendix B. (3) TextMixer has only been evaluated at the BERT-like model and has not been migrated to larger models like GPT-3. Limited by computational resources, we can only use the pre-trained MIMO network provided by Murahari et al. (2023). However, we believe that service providers can train more powerful MIMO networks with more parameters and extend them to more tasks, such as text generation.

## Acknowledgements

The authors wish to thank the anonymous reviewers for their helpful comments. This work was partially funded by National Natural Science Foundation of China (No.62206057,62076069,61976056), Shanghai Rising-Star Program (23QA1400200), Shanghai Academic Research Leader Program 22XD1401100, and Natural Science Foundation of Shanghai (23ZR1403500).

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

## A    Datasets

we conduct experiments on 4 classification tasks and the statistics of datasets in our experiments are shown in Table 5. For all the datasets, we use the test data for evaluation. We illustrate the details of each dataset as follows:

**IMDB** (Maas et al., 2011) contains a large collection of movie reviews along with their corresponding sentiment labels, indicating whether the review is positive or negative.

**AGNEWS** (Zhang et al., 2015) provides a balanced distribution of articles across the World, Sports, Business, and Science/Technology, making it suitable for training and evaluating text classification models.

**CoNLL2003** (Tjong Kim Sang and De Meulder, 2003) is a popular benchmark dataset for named entity recognition (NER) tasks. It consists of English and German news articles annotated with named entity labels such as person names, locations, organizations, and miscellaneous entities.

**OntoNotes5** (Weischedel et al., 2013) contains text from various domains, including news, conversations, and web data, and is annotated with detailed named entity labels such as person names, locations, organizations, and more.

| Dataset | # Train | #Test | #Labels | #Average Length |
|---------|---------|-------|---------|-----------------|
| IMDB | 25000 | 25000 | 2 | 272.26 |
| AGNEWS | 120000 | 7600 | 4 | 51.80 |
| CoNLL2003 | 14041 | 3453 | 9 | 19.21 |
| OntoNotes5 | 59924 | 8262 | 37 | 21.19 |

Table 5: Statistics of the datasets.

## B    Privacy Analysis

Given a real input $\mathbf{X} = [\mathbf{x}_1, \mathbf{x}_2, ..., \mathbf{x}_n]$ with length $n$ and its synthetic inputs $\{\mathbf{X}_i\}_{i=1}^{k}$.

**Mixing Encryption**    Mixing encryption mixes $k$ word representations based on Eq. 1 and Eq. 2. There are a total of $k$ possible word representations (1 real + $k1$ synthetic). All representations are assumed to be equally likely. Besides that, The mixed representation is more confusing than the isolated representation, which further reduces the success rate of attacks. Hence, the probability of correctly guessing the real input is less than the reciprocal of the total number of inputs, $P_{(Mix\_Attack)} \leq \frac{1}{k}$.

**Representation Encryption**    Representation encryption adds random noise to a random representation before mixing. "This encryption method draws on differential privacy, which increases the level of confusion in the representation. Although it is difficult to analyze the theoretical privacy protection capability, we provide empirical results on the impact of different noise scales on privacy and performance in Appendix E.

**Position Encryption**    Given a shuffled sentence of length $n$, there are $n!$ (n factorial) ways to arrange $n$ items. All arrangements are assumed to be equally likely. Hence, the probability of correctly guessing the original order should be the reciprocal of the total number of arrangements: $P_{(Pos\_Attack)} = \frac{1}{n!}$. Although attackers can exploit rules such as grammar and positional information to eliminate and increase the success rate of attacks, there would still be many confusing candidates left. We also designed privacy attacks in §4.2 to verify this.

## C    Discussion about TextFusion

TextFusion (Zhou et al., 2022) aims to fuse token representations and prevent privacy attackers from collecting ideal data and training a one-to-one inversion model. In their privacy setting, privacy attackers come primarily from third parties. However, our setting is stricter than TextFusion, service providers can also be potential privacy attackers, they release the model weights, can design special attack methods, and collect any training data if they want. We propose a simple attack method and use it in our experiments. For token-level tasks, we train an inversion model that only restores unfused tokens and directly uses this model to attack. For sentence-level tasks, we train an inversion to restore one original word from fused representations. As shown in Table 1, the success rates of both attack methods are considerably high, indicating that our scenario necessitates stronger methods for privacy protection.

## D    Discussion about Position Encryption

In §2.3, we mention that $\mathbf{X}_{PE}^{mix}$ and $\mathbf{X}^{mix}$ are equivalent for PLM, which means that even if the word positions are shuffled, it does not affect the subsequent model inference. Here we provide a brief explanation, we use a Bert-like model, and no relative position embedding is used. As a

| Num. | $\epsilon$ | Task ↑ | Privacy ↓ |
|---|---|---|---|
| 2 | 1 | 89.5 | 64.1 |
| | 0.5 | 88.22 | 32.24 |
| | 0.2 | 81.29 | 4.48 |
| 5 | 1 | 87.15 | 6.96 |
| | 0.5 | 86.65 | 3.94 |
| | 0.2 | 83.32 | 1.62 |
| 10 | 1 | 84.39 | 1.64 |
| | 0.5 | 83.49 | 1.57 |
| | 0.2 | 81.91 | 1.26 |

Table 6: Effects of mixed inputs number and noise scale in CoNLL03. "Num." denotes the number of mixed inputs. "$\epsilon$" represents the noise parameter. A smaller value of $\epsilon$ corresponds to a larger noise scale. "Task" means task performance, specifically the F1 score for CoNLL03. "Privacy" refers to the privacy metric, which is the Top-1 accuracy of the Inversion attack. ↑ indicates that higher values are better, while ↓ suggests that lower values are better.

result, the position information is only provided by position embedding, which is added to each input before mixing. During the inference phase, the subsequent calculations (self-attention, softmax, feed-forward layer, etc.) are independent of the explicit representation position and only depend on the implicit positional features within the representation. As a result, we can shuffle the mixed representations without any impact on performance. However, the shuffled word positions can reduce the readability for humans, which confuses privacy attackers and enhances privacy.

## E    Effect of Mixed Inputs Number and Noise Scale

In this section, we show the utility-privacy trade-off caused by the number of mixed inputs and noise scale. We conducted experiments on CoNLL03, and Table 6 shows the experimental results for mixed inputs numbers of $[2, 5, 10]$ and noise parameters $\epsilon$ of $[1, 0.5, 0.2]$. lower $\epsilon$ means larger noise. It can be observed that increasing the noise scale appropriately can significantly prevent privacy leakage (64.1 -> 32.24 for N=2) with an acceptable decrease in performance (89.5 -> 88.22 for N=2). In addition to this, we can observe that incorporating mixed inputs results in a more favorable balance between utility and privacy, underscoring the significance of mixing.

| Method | FLOPS (User) | FLOPS (Server) | Inference Time | Communication Cost |
|---|---|---|---|---|
| Fine-tuning (Devlin et al., 2018) | 0.0003G | 10.88G | 0.077 s | 384 KB |
| PUMA* (Dong et al., 2023) | - | - | 33.91 s | 10.77GB |
| TextFusion (Zhou et al., 2022) | 0.0075G | 5.45G | 0.047 s | 192KB |
| TextMixer (N=5) | 0.1080G | 10.88G | 0.079 s | 384KB |
| TextMixer (N=10) | 0.2157G | 10.88G | 0.089 s | 384KB |

Table 7: Inference cost of different privacy-preserving methods. FLOPs stands for floating point operations. Inference Time represents the total inference time of the entire model, measured in seconds. N represents the number of mixed words. * denotes results are directly taken from (Dong et al., 2023).

## F Comparison of Inference Cost

To show the efficiency of TextMixer, we compare the inference cost of different types of privacy-preserving methods, including Fine-tuning, encryption-based methods, TextFusion and TextMixer. The communication cost is how much data the users send. Because inference time can vary depending on different platforms and implementations, we use both inference time and FLOPs (floating-point operations) as two metrics to measure computational costs. We use a sentence of length 128 to query a bert-base model, all experiments are conducted on the same CPU unless otherwise specified.

From Table 7, we can find that TextMixer only introduces a few additional computation and communication costs compared to fine-tuning. The fast encryption-based method, PUMA, still requires 33.91 seconds and 10.77GB communication cost, which is not convenient in real applications.

## G Implementation Details

The pre-trained MIMO network is released by (Murahari et al., 2023)[3], BERT-base are used as pre-trained MIMO network to implement TextMixer. To ensure consistency, all baseline methods are implemented using the BERT-base model. All the experiments are conducted on NVIDIA GeForce RTX 3090. Implementation details and the Hyper-Parameters of both attack and defense methods are introduced as follows.

**Defence Methods** Our implementation is based on the Hugging Face Transformer models [4] and is replication with publicly available code. At training time, we use the AdamW optimizer

(Loshchilov and Hutter, 2019) and a linear learning rate scheduler, as suggested by the Hugging Face default setup. For all defense methods, we choose Laplace noise to disturb the representation and train 30 epochs to guarantee convergence. We prioritize privacy while considering performance when selecting the most favorable outcomes from the experimental results for reporting. The optimized hyperparameters, which yielded the best results, are presented in Table 8.

For **TextMixer**, the hyperparameters we tune include the number of mixing instances and the scale of Laplace noise of Representation Encryption. We train TextMixer with the number of instances $N$ in [2, 5, 10] and the scale of Laplace noise $\epsilon$ in [1, 2, 2.5, 5, 6, 7, 8, 10]. For all datasets, we use Real Data as the synthetic inputs due to the better privacy-performance trade-off.

For **DPNR**[5], we take the scale of Laplace noise and the nullification rate for word dropout strategy as hyperparameters, we search the scale of the noise $\epsilon$ in [0.2, 1, 2, 10, 20] and the nullification rate $nu$ in [0, 0.1, 0.3, 0.5].

For **CAPE**[6], during the progress of adversarial training weight, the coefficient $\lambda_{adv}$ for the adversarial training loss is searched within the range of [0.01, 0.05, 0.1, 0.5, 1, 5] and the scale of the noise $\epsilon$ in [0.2, 1, 2, 10, 20].

For **Santext+**[7], our approach aligns with the author's methodology, utilizing GloVe (Pennington et al., 2014) for word substitution guidance. By default, the probability of non-sensitive words being replaced is set at 0.3 (denoted as $p$), while the percentage of sensitive words to be sanitized is set at 0.9 (denoted as $w$). We conduct a search for privacy parameter $\alpha$ within the range of [1, 2, 3].

---

[3]https://github.com/princeton-nlp/datamux-pretraining/
[4]https://huggingface.co/

[5]https://github.com/xlhex/dpnlp
[6]https://github.com/NapierNLP/CAPE
[7]https://github.com/xiangyue9607/SanText

For **DataMix**, we implement the method ourselves based on the original paper, we search the number of mixing instances in the range of [2, 4, 8].

For **TextFusion**, we focus on adjusting the misleading weight during the fusion process. we vary the misleading weight $\lambda_{ml}$ within the range of [0.05, 0.5, 1, 5, 10, 15].To enhance privacy on sentence-level tasks, we increase the weight of the misleading loss term.

**Attack Methods**   To carry out the **KNN Attack** in our implementation, we utilized the embedding matrix of the BERT-base model. This embedding matrix was employed to compute the Euclidean distance between the client representations. In the case of **InvBert Attack and MLC Attack**, we employ the BERT-base model to construct the inversion model and train 20 epochs to guarantee convergence. We also perform a search for the optimal learning rate within the range of [1e-4, 1e-5, 2e-5, 1e-6]. For baselines, we use a learning rate of 2e-5 to train the inversion model. For TextMixer, the best learning rate we tuned is 1e-4 in most cases.

| | bsz | lr | $N$ | $\epsilon$ | $nu$ | $\lambda_{adv}$ | $\alpha$ | $\lambda_{ml}$ |
|---|---|---|---|---|---|---|---|---|
| IMDB: | | | | | | | | |
| Fine-tuning | 32 | 2e-5 | - | - | - | - | - | - |
| DPNR | 32 | 2e-5 | - | 5 | 0 | - | - | - |
| CAPE | 32 | 5e-5 | - | 10 | - | 1 | - | - |
| SanText+ | 64 | 1e-5 | - | - | - | - | 3 | - |
| DataMix | 32 | 2e-5 | 4 | - | - | - | - | - |
| TextFusion | 32 | 5e-5 | - | - | - | - | - | 15 |
| TextMixer | 32 | 5e-5 | 5 | 0.4 | - | - | - | - |
| AGNEWS: | | | | | | | | |
| Fine-tuning | 32 | 2e-5 | - | - | - | - | - | - |
| DPNR | 32 | 2e-5 | - | 1 | 0 | - | - | - |
| CAPE | 32 | 5e-5 | - | 2 | - | 0.1 | - | - |
| SanText+ | 64 | 1e-5 | - | - | - | - | 1 | - |
| DataMix | 32 | 2e-5 | 2 | - | - | - | - | - |
| TextFusion | 32 | 5e-5 | - | - | - | - | - | 15 |
| TextMixer | 32 | 5e-5 | 10 | 6 | - | - | - | - |
| CoNLL2003: | | | | | | | | |
| Fine-tuning | 32 | 2e-5 | - | - | - | - | - | - |
| DPNR | 32 | 2e-5 | - | 1 | 0 | - | - | - |
| CAPE | 32 | 5e-5 | - | 5 | - | 0.1 | - | - |
| SanText+ | 64 | 1e-5 | - | - | - | - | 3 | - |
| DataMix | 32 | 2e-5 | 2 | - | - | - | - | - |
| TextFusion | 32 | 5e-5 | - | - | - | - | - | 1 |
| TextMixer | 32 | 5e-5 | 5 | 0.5 | - | - | - | - |
| OntoNotes5: | | | | | | | | |
| Fine-tuning | 32 | 2e-5 | - | - | - | - | - | - |
| DPNR | 32 | 2e-5 | - | 1 | 0 | - | - | - |
| CAPE | 32 | 5e-5 | - | 5 | - | 0.1 | - | - |
| SanText+ | 64 | 1e-5 | - | - | - | - | 3 | - |
| DataMix | 32 | 2e-5 | 4 | - | - | - | - | - |
| TextFusion | 32 | 5e-5 | - | - | - | - | - | 1 |
| TextMixer | 32 | 5e-5 | 5 | 0.4 | - | - | - | - |

Table 8: Hyperparameter settings for our method and baseline methods. - represents the hyperparameter is not used in this method. $N$ is the number of mixing instances used in TextMixer and DataMix. $\epsilon$ is the noise scale used in TextMixer, DPNR and CAPE. $nu$ is the nullification rate of word dropout for DPNR. $\lambda_{adv}$ is the weight of the adversarial training loss for CAPE. $\alpha$ is the privacy parameter for Santext+. $\lambda_{ml}$ is the misleading loss weight used in TextFusion.