# OpenReview forum: "TextMixer: Mixing Multiple Inputs for Privacy-Preserving Inference"
_EMNLP/2023/Conference — EMNLP 2023 Findings_

### Official Review · Reviewer_qUde · 2023-08-04

**Soundness:** 4

**Excitement:**

3: Ambivalent: It has merits (e.g., it reports state-of-the-art results, the idea is nice), but there are key weaknesses (e.g., it describes incremental work), and it can significantly benefit from another round of revision. However, I won't object to accepting it if my co-reviewers champion it.

**Paper Topic And Main Contributions:**

This paper is about protecting user's private data that may be sent over to Large Language Models deployed in cloud services. Even in the case user information is converted to corresponding embeddings before sending over to the cloud based LLM, they are still susceptible to embedding inversion attacks. The major contributions are the following:
* It comes up with a technique that uses mixing-centric encryption method, along with mixing private input with synthetically generated artificial input. This approach is shown to preserve user privacy under certain attacks.
* It further ensures the model performance is not affected too much, which is typically a tradeoff in privacy preserving systems.
* It comes up with a technique to generate synthetic inputs that help with the mixing techniques described in the paper.

**Reasons To Accept:**

The paper addresses a very pertinent issue of protection of private user data that is being sent to a cloud service for use with a Large Language Model. The major strength of the paper are the following:
* The technique described is able to minimize privacy leakage under different attacks, and still maintain comparable model performance. It uses mixing-centric encryption method and adds multiple obfuscation techniques using synthetically generated data.
* The mixing-centric encryption described here, is an innovative use of the MIMO network described by Murahari et al., 2023.
* It provides techniques for synthetic data generation that can be used with this technique, and also compares performance of the techniques.

**Reasons To Reject:**

The major weaknesses of this paper are the following:
* While the technique described shows privacy protection in the experiments, it doesn't provide any proof or bounds of privacy when the technique is applied to other datasets, and other models.
* The technique described is only applicable to text data. The synthetic data generation technique also has similar limitations.
* The technique would be computationally expensive on the user side.

**Reproducibility:**

3: Could reproduce the results with some difficulty. The settings of parameters are underspecified or subjectively determined; the training/evaluation data are not widely available.

**Reviewer Confidence:**

3: Pretty sure, but there's a chance I missed something. Although I have a good feel for this area in general, I did not carefully check the paper's details, e.g., the math, experimental design, or novelty.

**Typos Grammar Style And Presentation Improvements:**

* Lap(\epsilon) is used in Eq. 3. It'd be great if you can please describe \epsilon.
* The paper refers to vanilla mixing with reference to the MIMO technique from Murahari et al, 2023. It would be great if there is a brief explanation of vanilla mixing added when it is mentioned the first time.
* Section 3.1 Datasets - Second sentence - "For sentence classification task..." would need to be rephrased.
* Section 4.2 Ablation Study - You mention that adding appropriate scale noise can enhance privacy without compromising much performance. It would be good to validate this hypothesis or mention the reason you think that is the case, as a lot of work on Differential Privacy shows that adding noise this way does affect performance.

---

> ### Author Rebuttal · Authors · 2023-08-29
>
> Thanks for your valuable comments!
>
> # A. Question about proof of privacy.
>
> 1. We provide an intuitive and simplified proof of privacy below, and **we plan to include a formal privacy proof in Appendix D of future versions**. As mentioned in the Limitations section, the privacy proof of MixPi is not as rigorous as that of homomorphic encryption.
>     - **Mixing Encryption**: If we mix the real input with k-1 synthesize inputs, the **privacy attackers will only have a $\frac{1}{k}$ probability of guessing the real input correctly**. Moreover, MixPi provides a single mixed representation instead of a set of k representations, which further increases the difficulty of privacy attacks.
>     - **Position Encryption**: If we shuffle a sentence of length $n$, $n$ random words can form $n!$ different sentences, so the **privacy attackers has a $\frac{1}{n!}$! chance of recovering the true order**. Although they can eliminate some candidate sentences through grammar, there would still be many confusing candidates left.
>
> 2. MixPi is dataset-agnostic and can be generalized to any dataset. Additionally, our approach could potentially be extended to the majority of Transformer-based models. The following are the conditions that different encryption methods need to satisfy:
>     - **Mixing Encryption:** This requires the model to support Multiple-Input Multiple-Output (MIMO), which can be achieved through pre-training [1,2].
>     - **Representation Encryption:** This method is model-agnostic, meaning it can be applied to any model.
>     - **Position Encryption:** For this method, the model needs to satisfy the shuffle equivalence property (details can be found in Appendix B). The majority of PLM architectures support this property.
>
>     If a model meets the requirements mentioned above, MixPi can be applied to it. This is entirely feasible, especially considering that the primary challenge appears to lie in MIMO pre-training for the specific modality, which might not be a significant issue for enterprises with ample computational resources.
>
>
> # B. Question about applications beyond textual data.
>
> This is a great question! We believe MixPi has the potential to be extended to other modalities. Currently, as the open-source MIMO model only supports the text modality [1,2], we can only conduct experiments on text data. Fortunately, our method has the potential to be extended to other Transformer-based models, and transformer-based models are widely used across various modalities. Therefore, if there are models in other modalities that meet the requirements outlined in Response A.2, we believe that it would only require minor adjustments to extend MixPi to those corresponding modalities (such as images and audio).  Thank you for your inspiration, in our future work, we will attempt to create a unified version of MixPi that works across multiple modalities!
>
> # C. Question about the computational cost of the users’ side.
>
> MixPi will not impose significant computational costs on the user's side. Below are some of our preliminary results on computational costs.
>
> We use a sentence of length 128 to query a bert-base model. All experiments are conducted on the same CPU since most cryptographic-based methods do not support GPU.  FLOPs stands for floating point operations. N represents the number of mixed words.
>
> | Method | FLOPS (User) | FLOPs (Server) |  |  |
> | --- | --- | --- | --- | --- |
> | Fine-tune | 0.0003G | 10.88G |  |  |
> | MixPi (N=5) | 0.1080G | 10.88G |  |  |
> | MixPi (N=10) | 0.2157G | 10.88G |  |  |
>
> From the Table, we can observe that **MixPi only introduces a few additional computational costs compared to fine-tuning (directly sending word embeddings)**, this computational cost is very small compared to the computation required for cloud PLMs. Given the current advancements in smart chips. We believe that with the development of modern chips, this amount of computation is within an acceptable range for users.
>
> **In addition to these preliminary results, we plan to add Appendix E for a detailed comparison of communication and computation costs among Fine-tuning, cryptographic-based methods, TextFusion, and MixPi.**
>
> # D. Suggestions about typos and presentation.
>
> Thank you for your thorough review! Based on your feedback, we will make the following changes:
> - Add a description of ε after Eq. 3, where $\epsilon$ is a hyperparameter controlling the scale of noise – smaller values of ε correspond to a larger scale of noise.
> - Provide a brief description of vanilla mixing at the end of line 143.
> - Rewrite Section 3.1 for improved readability. -
> - Regarding the problem of noise scale. We appreciate your reminder, our statement might be somewhat misleading. In fact, our findings are consistent with prior works, and we also agree that adding noise does indeed impact performance. Our point is that the appropriate noise would not result in **much** performance loss. This is acceptable as it contributes largely to privacy enhancement. As shown in Table 2, removing noise (- Repr) generally leads to improved task performance (92.32->92.63 in IMDB and 86.24->86.69 in CoNLL03), but privacy leakage becomes worse (2.22->6.57 in IMDB and 4.14->15.53 in CoNLL03). **We will revise this potentially misleading statement** and emphasize that noise does indeed affect performance, but these performance declines are acceptable in some cases.
>
> **If our responses have addressed your concerns about this paper, could you please raise your scores?**
>
> [1] DataMUX: Data Multiplexing for Neural Networks. NeurIPS 2022
>
> [2] MUX-PLMs: Data Multiplexing for High-throughput Language Models.

---

### Official Review · Reviewer_ekHf · 2023-08-05

**Soundness:** 4

**Excitement:**

4: Strong: This paper deepens the understanding of some phenomenon or lowers the barriers to an existing research direction.

**Paper Topic And Main Contributions:**

The paper proposes a privacy-preserving inference framework inspired by k-anonymity. The experiment results show promising improvement in terms of communication cost and privacy protection.

**Reasons To Accept:**

- Interesting work, with strong justification for its motivations.

- The proposed framework demonstrates a reasonable performance and lower privacy leakage compared to prior studies.

- The paper is well written


**Reasons To Reject:**

The reviewer has concerns regarding the usability of the privacy-preserving inference framework.  In section 5 (lines 540-543), the author mentioned that cryptographic approaches suffer from very high communication costs. Do the authors plan to evaluate the communication cost of deploying the proposed framework and make a comparison between cryptographic approaches and similar approaches (e.g., TextFusion)?

**Reproducibility:**

3: Could reproduce the results with some difficulty. The settings of parameters are underspecified or subjectively determined; the training/evaluation data are not widely available.

**Reviewer Confidence:**

2: Willing to defend my evaluation, but it is fairly likely that I missed some details, didn't understand some central points, or can't be sure about the novelty of the work.

---

> ### Author Rebuttal · Authors · 2023-08-29
>
> Thank you for your suggestion!
>
> We do plan to add comparisons of communication costs across different methods. Below are some of our preliminary results on communication costs.
>
> We use a sentence of length 128 to query a bert-base model. All experiments, except for PUMA, are conducted on the same CPU since most cryptographic-based methods do not support GPU. * denotes results are directly taken from [1]. N represents the number of mixed words.
>
> | Method | Communication Cost |  |  |  |
> | --- | --- | --- | --- | --- |
> | Fine-tune | 384 KB  |  |  |  |
> | PUMA (Cryptographic-Based)* [1] | 10.773 GB |  |  |  |
> | TextFusion [2] | 192KB |  |  |  |
> | MixPi (N=5) | 384 KB  |  |  |  |
> | MixPi (N=10) | 384 KB  |  |  |  |
>
> From the table, we can see that **the communication cost of MixPi is the same as Fine-tune** (sending original word representations directly). This is because our method only requires users to send encrypted word representations to the cloud (we only count the communication costs for data sent from users).  TextFusion sends fused word representations to the cloud. Token fusion reduces the sentence length, so its communication cost is less than MixPi and Fine-tune. Cryptographic-based methods have much higher communication costs than our method due to the need to support extensive cryptographic protocols.
>
> **In addition to these preliminary results, we plan to add Appendix E for a detailed comparison of communication and computation costs among Fine-tuning, encryption-based methods, TextFusion, and MixPi.**
>
> [1] PUMA: Secure Inference of LLaMA-7B in Five Minutes. Arxiv 2023
>
> [2] TextFusion: Privacy-Preserving Pre-trained Model Inference via Token Fusion. EMNLP 2022

---

### Official Review · Reviewer_hBPv · 2023-08-05

**Paper Topic And Main Contributions:** The paper proposes a privacy-preservi…
**Soundness:** 1

**Excitement:**

2: Mediocre: This paper makes marginal contributions (vs non-contemporaneous work), so I would rather not see it in the conference.

**Missing References:**

Carlini N, Deng S, Garg S, et al. Is private learning possible with instance encoding?[C]//2021 IEEE Symposium on Security and Privacy (SP). IEEE, 2021: 410-427.

**Reasons To Accept:**

1) The privacy-preserving inference in language model is emerging issue.
2) The paper clearly presents the methodology with a detailed diagram and easy to follow.

**Reasons To Reject:**

1) The privacy guarantee lacks rigorous theoretic proof. The paper is to propose a complex and advanced method compared with the previous TextHide with perturbations and shuffle.
2) The k-anonymity here is a weaker privacy notion, which could be broken potentially.
3) It would be ideal if the paper could have a lightweight inference while preserving privacy. The paper should include utility-privacy analysis .

**Reproducibility:**

3: Could reproduce the results with some difficulty. The settings of parameters are underspecified or subjectively determined; the training/evaluation data are not widely available.

**Reviewer Confidence:**

4: Quite sure. I tried to check the important points carefully. It's unlikely, though conceivable, that I missed something that should affect my ratings.

---

> ### Author Rebuttal · Authors · 2023-08-28
>
> Thanks for your valuable comments.
> # A. Misunderstanding of privacy scenarios.
>
> Based on your “……a complex and advanced method compared with the previous TextHide……” and the Missing References you mentioned, we are guessing you misunderstood that our privacy scenario is similar to TextHide's. But in fact, **their privacy scenarios are completely different. Moreover, the attack methods for TextHide\&InstaHide [4] are not applicable to our scenarios and approach.**
>
>
> 1. **The privacy scenarios for TextHide and MixPi are totally different,** as we mentioned in the Related Work section (lines 558-563). TextHide\&InstaHide are proposed for "**private** **learning**", but MixPi is proposed for "**private** **inference**".  [3] has discussed the difference between various privacy scenarios and attack methods.
>     - **In TextHide's privacy scenario, the privacy risk comes from the training phase**. It aims to prevent the training data from leaking to other parties during distributed or federated learning. **TextHide protects the privacy of training set** and it aims to defend training data attack such as membership attacks [1].
>     - **In our privacy scenario, the privacy risk comes from the inference phase**. More specifically, **after training**, some fine-tuned PLMs will be deployed on the cloud and released as cloud inference services. Users can query these inference services with their personal data to perform target NLP tasks. Our method aims to allow users query inference services without leaking private data to service providers or third parties. **MixPi can be seen as protecting the privacy of "test set"** and it aims to defends privacy attack for inference phase, such as embedding inversion attacks [2, 3].
>     - **The capabilities and objectives of the privacy attacker are different in these two different privacy scenarios**. For example, [4] assume attackers is given access to the encoded training dataset $\hat{S}$,  their objective  is to recover training text $x_{train}$ in the original dataset $S$ as much as possible. But in our scenario,  the attack is given access to a single encrypted embeddings $H$, their objective is to recover the original sentence $x_{test}$, and this sentence may be  just generated by the user and does not exist in the training data. Therefore, the privacy attack for TextHide’s privacy scenario is not applicable to our privacy inference scenario. And **the method and conclusion drawn from *Is private learning possible with instance encoding* [4],** which proposes a specific attack method for InstaHide,  **are not applicable to our scenario and approach. It  also cannot prove our method can be easily broken.**
>
> # **B. Concerns about privacy guarantee.**
>
> Firstly, MixPi’s privacy guarantees can be theoretically proven. Secondly, we have empirically verified MixPi's privacy through rigorous experiments. Thirdly, we believe empirical method with extensive experimentation can also make sufficient contributions.
>
>
>
> 1. **The privacy guarantees of MixPi can be theoretically proven.** We mentioned in the limitations section that MixPi’s proof is not as rigorous as homomorphic encryption, the following is a simple, intuitive proof.
>     - **Mixing Encryption**: If we mix the real input with k-1 synthesize inputs, the **privacy attackers will only have a $\frac{1}{k}$ probability of guessing the real input correctly**. Moreover, MixPi provides a single mixed representation instead of a set of $k$ representations, which further increases the difficulty of privacy attack.
>     - **Position Encryption**: If we shuffle a sentence of length $n$, $n$ random words can form $n!$ different sentences, so the **privacy attackers have a $\frac{1}{n!}$ chance of recovering the true order**. Although they can eliminate some candidate sentences through grammar, there would still be many confusing candidates left.
>
>     We will add proof of the privacy guarantee in Appendix D to enhance the quality of our paper.
>
> 2. **The privacy guarantees of MixPi has been empirically validated**.
>     - **We have conducted rigorous experiments to validate MixPi's ability to protect privacy.** We leveraged the most powerful and widely used privacy attack methods (Embedding Inversion Attack [2, 3, 6])  to attack MixPi (Section 4.1). We also designed specialized attack methods for different encryption modules of MixPi (Section 4.3). The experimental results demonstrate that MixPi can effectively resist general and specialized privacy attacks, and outperforms methods with formal privacy proofs (e.g. SanText) in both privacy and performance.
>     - We believe that the strength of privacy notion is determined by the privacy scenario, and whether a method can be broken depends on the capabilities of potential attackers. This is why simple and intuitive techniques like k-anonymity are still widely employed by companies like Google. In our scenario, based on the theoretical proof and empirical experiments, we believe that MixPi's privacy guarantees are sufficient to effectively defend privacy attacks. Of course, we are also very willing to compare with more strong attack methods that may appear in the future and believe that our method can also achieve good results in these attack methods : ).
> 3. It's worth mentioning that even without strict privacy guarantees, some empirical methods can still protect privacy under their settings, such as [6] and [7]. From a broader perspective, many works only propose the empirical methods and comprehensive experiments without rigorous proofs, but these works have still made significant contributions, such as GPT [9], Mixup [10] and adversarial training [11].  **We believe that our work, which propose empirical method and conducts comprehensive experiments, has made enough contribution.** Nonetheless,  we are also plan to further improve our paper by adding a theoretical proof section as you suggest.
>
> # C.Mixing inputs for privacy during inference faces new challenges.
>
>
> Our approach is not simply "a complex and advanced method compared with the previous TextHide". TextHide only mixes inputs in the training phase and it uses one input for inference without any mixing. In contrast, MixPi aims to mix multiple inputs during the inference phase while preserving task performance, which presents new challenges. For example, if two inputs, $x_1$ and $x_2$, are mixed using a TextHide style like $x' = 0.3 * x_1 + 0.7 * x_2$, the resulting mixed input $x'$ fed into the model for inference will yield a "mixed" prediction $y'$. Even with coefficients 0.3 and 0.7, it's challenging to directly extract the desired predictions, $y_1$ or $y_2$, from mixed prediction $y'$. Furthermore, as the number of mixed inputs increases, even if the model is trained on the mixed data, it is still difficult for the model to handle the mixed input during inference, leading to poor performance. A private inference method for images, DataMix [2], has a mixing method similar to TextHide. However, as shown in Table 1, the performance and privacy of DataMix are very poor, indicating that these challenges cannot be addressed easily. **MixPi overcome the above new challenges. Therefore, our method is not just an extension of TextHide, but rather establishes a novel framework that mixes multiple inputs to achieve privacy-preserving inference**.
>
> # **D. Concerns about lightweight&utility-privacy analysis.**
>
> We are not very certain about what specific concern you have, but our method is indeed lightweight, and we provide a significant amount of utility-privacy analysis in Section 4.
>
> 1. Our framework is very lightweight for users. Users only need to convert text into word representations and encrypt these word representations in their local devices. The main computations are performed by the cloud services. Because we haven't encrypted the computations of the model, the **inference time per input of MixPi is similar to the normal, original PLM**.
> 2. Below are some preliminary results. We use a sentence of length 128 to query a bert-base model. All experiments, except for PUMA, are conducted on the same CPU since most cryptographic-based methods do not support GPU. * denotes results directly taken from [12].  FLOPs stands for floating point operations. Inference Time represents the total inference time of the entire model, measured in seconds. N represents the number of mixed words.
>
>
>
>     | Method | FLOPS (User) | FLOPs (Server) | Inference Time |  |
>     | --- | --- | --- | --- | --- |
>     | Fine-tune | 0.0003G | 10.88G | 0.077 s |  |
>     | PUMA (Cryptographic-Based)* [12] | \ | \ | 33.91 s |  |
>     | TextFusion [6] | 0.0075G | 5.45G | 0.047 s |  |
>     | MixPi (N=5) | 0.1080G | 10.88G | 0.079 s |  |
>     | MixPi (N=10) | 0.2157G | 10.88G | 0.089 s |  |
>
>     In addition to these preliminary results, we plan to provide a more detailed comparisons of the communication costs and computational costs of Fine-tune, encryption-based methods, TextFusion, and MixPi in Appendix E.
>
> 3. We show the main utility-privacy analysis in Section 4. We use task performance as the  utility metric in this paper. The task performance and privacy metric are shown in Table 1, the ablation study is shown in Table 2 and Figure 1, and the specific privacy attacks are shown in Table 3 and Table 4.  In addition, in order to enhance the quality of our paper, we also plan to add Appendix F, which is used to discuss the impact of the number of mixed inputs and noise scales on performance and privacy. Below are some preliminary results on CoNLL03.
>
>
>     | Number of mixed inputs | Noise Scale (eps) | Task Metric (F1) | Privacy Metric (Inverison top1) |
>     | --- | --- | --- | --- |
>     | 2 | 1 | 89.50 | 64.17 |
>     | 2 | 0.5 | 88.22 | 32.24 |
>     | 2 | 0.2 | 81.29 | 4.48 |
>     | 5 | 1 | 87.15 | 6.96 |
>     | 5 | 0.5 | 86.65 | 3.94 |
>     | 5 | 0.2 | 83.32 | 1.62 |
>     | 10 | 1 | 84.39 | 1.64 |
>     | 10 | 0.5 | 83.49 | 1.57 |
>     | 10 | 0.2 | 81.91 | 1.26 |
>
>
> **Based on the above response, we propose a novel privacy-preserving inference framework with theoretical and empirical privacy guarantees. We believe our experiments are sound, contributions are enough and deserve higher scores for soundness and excitement.**
>
> [1] Invernet: An Inversion Attack Framework to Infer Fine-Tuning Datasets through Word Embeddings. EMNLP2022
>
> [2] InvBERT: Reconstructing Text from Contextualized Word Embeddings by inverting the BERT pipeline.
>
> [3] Information Leakage in Embedding Models. CCS 2020
>
> [4] Is private learning possible with instance encoding?  2021 IEEE Symposium on Security and Privacy. IEEE Symposium on Security&Privacy 2021
>
> [5] DataMix: Efficient Privacy-Preserving Edge-Cloud Inference. ECCV 2020
>
> [6] TextFusion: Privacy-Preserving Pre-trained Model Inference via Token Fusion. EMNLP 2022
>
> [7] Differential Privacy for Text Analytics via Natural Text Sanitization. ACL 2021
>
> [8] CAPE: Context-Aware Private Embeddings for Private Language Learning. EMNLP 2021
>
> [9] Language Models are Few-Shot Learners. 2020
>
> [10] mixup: Beyond Empirical Risk Minimization. ICLR 2018
>
> [11] Explaining and Harnessing Adversarial Examples. ICLR 2015
>
> [12] PUMA: Secure Inference of LLaMA-7B in Five Minutes. Arxiv 2023

---

### Meta-Review · Area_Chair_6AY3 · 2023-09-24

**Recommendation:** 4

**Metareview:**

The paper addresses an emerging and vital issue in PLM. I agree with the reviewers ekhf and qUde. It addresses a pertinent issue in the field of data privacy with LLMs and presents an innovative privacy-preserving technique that performs well. The paper's clarity and the inclusion of techniques for synthetic data generation further enhance its value.

---

### Decision · Program_Chairs · 2023-10-07

**Decision:**

Accept-Findings

**Comment:**

The paper addresses an emerging and vital issue in PLM. I agree with the reviewers ekhf and qUde. It addresses a pertinent issue in the field of data privacy with LLMs and presents an innovative privacy-preserving technique that performs well. The paper's clarity and the inclusion of techniques for synthetic data generation further enhance its value.